# BRCA1 the Versatile Defender: Molecular to Environmental Perspectives

**DOI:** 10.3390/ijms241814276

**Published:** 2023-09-19

**Authors:** Amy X. Zhong, Yumay Chen, Phang-Lang Chen

**Affiliations:** 1Harvard-MIT Division of Health Sciences and Technology, Massachusetts Institute of Technology, Cambridge, MA 02139, USA; amyzhong@mit.edu; 2Department of Medicine, Division of Endocrinology, University of California, Irvine, CA 92697, USA; yumayc@uci.edu; 3Department of Biological Chemistry, University of California, Irvine, CA 92697, USA

**Keywords:** tumor suppressor gene, BRCA1, breast cancer, tissue specificity, ubiquitination, DNA damage repair, Bisphenol A, BRCT, RING finger, differentiation

## Abstract

The evolving history of BRCA1 research demonstrates the profound interconnectedness of a single protein within the web of crucial functions in human cells. Mutations in BRCA1, a tumor suppressor gene, have been linked to heightened breast and ovarian cancer risks. However, despite decades of extensive research, the mechanisms underlying BRCA1’s contribution to tissue-specific tumor development remain elusive. Nevertheless, much of the BRCA1 protein’s structure, function, and interactions has been elucidated. Individual regions of BRCA1 interact with numerous proteins to play roles in ubiquitination, transcription, cell checkpoints, and DNA damage repair. At a cellular scale, these BRCA1 functions coordinate tumor suppression, R-loop prevention, and cellular differentiation, all of which may contribute to BRCA1’s role in cancer tissue specificity. As research on BRCA1 and breast cancer continues to evolve, it will become increasingly evident that modern materials such as Bisphenol A should be examined for their relationship with DNA stability, cancer incidence, and chemotherapy. Overall, this review offers a comprehensive understanding of BRCA1’s many roles at a molecular, cellular, organismal, and environmental scale. We hope that the knowledge gathered here highlights both the necessity of BRCA1 research and the potential for novel strategies to prevent and treat cancer in individuals carrying BRCA1 mutations.

## 1. Introduction

Breast cancer remains a widespread affliction, with an estimated 297,790 new cases of invasive breast cancer projected in the United States in 2023 by the American Cancer Society. The seminal cloning of the breast susceptibility gene 1 (BRCA1) in 1994 heralded a significant advancement in the early diagnosis and understanding of breast cancer biology [1]. The BRCA1 gene product was identified as a nuclear phosphoprotein comprising 1863 amino acids [1,2,3], distinguished by the presence of structurally conserved elements at each flanking terminus. At the amino terminus, a distinct RING finger domain resides, recognized for its role in protein–protein interactions [4]. The BRCT domain, a self-contained folding unit defined by clusters of hydrophobic amino acids, further embellishes BRCA1’s structure. Intriguingly, this domain is shared by a diverse array of proteins implicated in DNA repair and cell cycle checkpoint control [5,6,7], a connection bolstered by earlier observations linking BRCA1 to DNA damage response.

Building upon these foundations, subsequent investigations further explored BRCA1’s interactions and associations. In 1997, Scully et al. reported a pivotal revelation—BRCA1 associates and co-localizes with Rad51, a recombinase akin to bacterial RecA, during the S phase in mitotic cells [8]. In 1999, our research extended these insights, highlighting BRCA1’s in vitro and in vivo interaction with hRad50, a key constituent of the hMre11-NBS1 complex [9]. This complex, identified as discrete foci within the nucleus, was dramatically reduced in HCC/1937 breast cancer cells bearing a homozygous BRCA1 mutation yet was restored via wild-type BRCA1 transfection. Collectively, these formative observations illuminate BRCA1’s vital role in orchestrating cellular responses to DNA damage, mediated by both Rad51 and the hRad50-hMre11-p95 complex. As the saga of BRCA1’s contributions unfolds, these early revelations anchor our journey toward unraveling the intricate tapestry of breast cancer biology.

Germline BRCA1 mutations confer a significant lifetime susceptibility to breast and ovarian cancer, with reported absolute risks of 45% to 87% for breast cancer and 36% to 66% for ovarian cancer by age 70 [1,10]. To further elucidate the impact on other cancer types, Thompson et al. conducted an extensive cohort study involving 11,847 individuals from 699 families across Europe and North America, revealing heightened risks of specific cancers among BRCA1 mutation carriers, such as pancreatic and uterine body/cervix cancers [11].

With three decades of extensive research efforts, significant advancements have been made in comprehending BRCA1’s subtleties, spanning from molecular to organismal levels. Our understanding of its tumor suppressor role, especially its enigmatic tissue specificity, has grown significantly. This review offers an overview of BRCA1’s molecular role as a tumor suppressor focusing on breast and ovarian cancers. It explores potential environmental factors as tissue mutagens that facilitate hormone-related tumorigenesis. A comprehensive understanding of BRCA1’s functions remains pivotal for formulating effective approaches in diagnosing, preventing, and treating breast and ovarian cancers. Identifying environmental factors that contribute to cancer development is equally imperative, as awareness is the basis of proactive prevention and risk mitigation.

## 2. BRCA1 Is Involved in Several Complexes

BRCA1 engages in various cellular processes via its diverse protein complexes (Figure 1). These complexes provide BRCA1 with the capacity to contribute to essential functions encompassing cellular differentiation, cell cycle regulation, DNA damage checkpoint responses, and DNA repair mechanisms. In the ensuing discussion, we will meticulously examine the most noteworthy complexes linked to different regions of the BRCA1 protein, beginning with the N-terminal RING finger, extending to the middle DNA binding and coiled-coil regions, and concluding with the C-terminal BRCT domain.

### 2.1. The BRCA1/BARD1 Complex Functions as a Ubiquitin Ligase

The distinctive N-terminal RING finger domain of the BRCA1 protein emerges as a pivotal determinant of its functionality, particularly within the context of the ubiquitination system. Ubiquitination, a precisely orchestrated process, engages a triad of enzymes: E1 (ubiquitin-activating enzyme), E2 (ubiquitin-conjugating enzyme), and E3 (ubiquitin ligase) [12]. This cascade coordinates the post-translational modification of target proteins by appending ubiquitin molecules, often flagging them for degradation or altering their roles. E1 primes ubiquitin molecules and transfers them to E2, which serves as a conveyor, transporting ubiquitin to the intended protein. E3 assumes a central role, dictating substrate specificity and facilitating the handover of ubiquitin from E2 to the target protein. This step is pivotal, as diverse E3 ligases engage distinct target proteins, ensuring the meticulous alteration of various cellular components. 

Within this framework, the BRCA1 RING finger domain facilitates interactions with a spectrum of proteins, including BARD1 [13]. Intriguingly, purified BRCA1 and BARD1 protein complexes exhibit auto-ubiquitin ligase activity, notably targeting ubiquitin K6 [14]. Moreover, in collaboration with specific E2s, these complexes catalyze the formation of K48- and K63-linked polyubiquitylated chains [15]. While many potential substrates have been posited for this activity [16], a thought-provoking revelation emerges: the E3 ligase function of BRCA1, as demonstrated in animal models, is not indispensably linked to its tumor suppression role [17].

This seemingly paradoxical observation assumes special significance, particularly in the context of human carriers of these mutations predisposed to familial breast cancer. It suggests an intriguing notion that the protein–protein interactions orchestrated by the RING finger domain fundamentally underlie its role in tumor suppression. Unraveling the exact implications of the BRCA1-BARD1 interplay in the landscape of breast cancer pathogenesis has the potential to reveal novel insights, potentially catalyzing the development of innovative therapeutic strategies.

### 2.2. The BRCA1/ZBRK1/CtIP Complex Functions as a Transcription Co-Repressor

The central region of BRCA1 has been established as having a connection to non-specific DNA binding activities. Masuda et al. have provided insight into the DNA binding region (DBR) (amino acids 421-701) of BRCA1, revealing its impact on genetic stability through active participation in the intra-S-phase checkpoint, particularly in response to replication stress [18]. 

While the DBR can directly bind DNA non-specifically, the DBR also demonstrates sequence-specific binding capabilities through intricate partnerships with various transcription factors, including ZBRK1, Myc, p53, and SP1 [19,20,21,22]. Of these transcription factors, ZBRK1 is particularly notable as a co-repressor. ZBRK1 consists of an N-terminal KRAB domain, followed by eight zinc fingers and a BRCA1-dependent transcriptional repression domain [23]. The final two zinc fingers and the C-terminal domain of ZBRK1 interlock with the central domain of BRCA1 (amino acids 341-748) to facilitate targeted recruitment to gene promoters [19]. Additionally, the DNA-binding capabilities of the ZBRK1 zinc fingers are contingent upon the binding of BRCA1 to the ZBRK1 C-terminal repression domain [23]. In this BRCA1/ZBRK1/DNA complex, BRCA1 also recruits histone deacetylase activities, underscoring the collaborative complexity of these interactions. 

Furthermore, ZBRK1 mediates a connection between BRCA1 and CtIP to participate in the transcriptional repression of Angiopoietin 1 and high mobility group AT-hook 2 [24,25]. This interaction between BRCA1, ZBRK1, and CtIP converges into a sophisticated complex that is assumed to be pivotal in suppressing critical genes for fundamental cellular processes, including cell cycle regulation, DNA repair, and apoptosis. 

Notably, recent research by Hong et al. has revealed a cluster of metabolic genes regulated by BRCA1 [26]. For example, the BRCA1-ZBRK1 complex coregulates the repression of GOT2. Disruption of this co-repressor complex results in upregulating GOT2 expression, subsequently triggering increased production of aspartate and alpha-ketoglutarate. This metabolic alteration fuels rapid cell proliferation in breast cancer cells. Notably, Hong et al. report that GOT2 can independently serve as a prognostic factor for overall and disease-free survival among breast cancer patients. This newfound role of BRCA1 in modulating GOT2 expression introduces yet another noteworthy facet to its role as a tumor suppressor. 

By deciphering the complex interplay between BRCA1 and its transcriptional co-regulators, we may uncover critical pathways that drive tumorigenesis and identify potential vulnerabilities in cancer cells. This knowledge could be instrumental in developing innovative therapies tailored to the specific molecular characteristics of breast and ovarian cancers associated with BRCA1 dysfunction. Ultimately, gaining a deeper understanding of BRCA1’s role as a transcriptional regulator may pave the way for improved cancer treatments and better outcomes for patients with BRCA1-related malignancies.

### 2.3. PALB2 Connects BRCA1 and BRCA2

The coiled-coil region of BRCA1 (amino acids 1369-1418) has been demonstrated to play a crucial role in recruiting BRCA2 to sites of DNA damage, achieved through its interaction with PALB2 (partner and localizer of BRCA2) [27]. As a scaffold protein, PALB2 facilitates the assembly of a functional complex involving both BRCA1 and BRCA2, a pivotal requirement for homologous recombination in DNA repair processes. Specifically, the N-terminal region of PALB2 binds to the coiled-coil domain of BRCA1, while the C-terminal WD-40 region of PALB2 interacts with the N-terminal region of BRCA2. This intricate binding arrangement forms a BRCA1-PALB2-BRCA2 complex, responsible for recruiting RAD51 to DNA damage sites, a fundamental step in homologous recombination repair mechanisms. 

Beyond merely acting as a bridge between BRCA1 and BRCA2, PALB2 significantly stabilizes these proteins and amplifies their effectiveness within DNA repair processes [27,28]. The importance of this complex’s biological function is exemplified by the fact that mutations in PALB2 have been directly associated with an elevated risk of breast cancer [29]. This underscores the pivotal role PALB2 plays in preserving genomic stability and thwarting cancer development. Thus, the coiled-coil region of BRCA1, through its interaction with PALB2, serves as a linchpin in coordinating the assembly of BRCA1, BRCA2, and RAD51 into a functional DNA repair complex. 

### 2.4. The BRCA1 A-, B-, C-Complex

The pivotal role of the C-terminal tandem BRCT domain of BRCA1 as a phosphopeptide-binding module is paramount, as it exerts a profound influence over DNA damage repair and the management of checkpoint control mechanisms [30,31,32]. This domain establishes vital interactions with phosphorylated Abraxas, Bach1, and CtIP, culminating in the assembly of the A-, B-, and C-complexes, respectively (reviewed in [33]). These complex formations are of utmost significance, encompassing diverse cellular processes that encompass DNA damage sensing, the propagation of critical signals, and the meticulous orchestration of DNA repair pathways. 

The BRCA1 A-complex, a deubiquitinating assembly composed of Rap80, Abraxas, NBA1, BRCC45, and BRCC36 [34,35,36]. Abraxas, in particular, serves as a central bridging molecule within this assembly. This interaction with the BRCT domain of BRCA1 is facilitated through the phosphorylated serine residue 400 of Abraxas. Within this complex, BRCC36 functions as a deubiquitinating enzyme with a specialized affinity for K63 polyubiquitin linkages [36]. This complex assumes a critical role in upholding genomic integrity amid cellular stress through its function as a deubiquitinating entity during the cellular response to DNA damage. 

To highlight the profound significance of the bridging molecule Abraxas in DNA damage repair and its potential role as a tumor suppressor, Castillo et al. conducted a comprehensive study involving Abraxas-deficient mice [37]. Surprisingly, the absence of Abraxas did not impede developmental processes, as post-birth, Abraxas-deficient mice displayed no discernible differences compared to their wild-type counterparts. However, the live Abraxas-deficient mice died between 7 and 28 days post-irradiation, in stark contrast to the uneventful survival of their wild-type counterparts within the same timeframe. Through meticulous investigations utilizing Abraxas -/- mouse embryonic fibroblasts (MEFs), Castillo et al. unraveled the critical functional domains of Abraxas in DNA damage repair. Notably, the absence of Abraxas led to a heightened susceptibility to lymphoma development, strongly indicating Abraxas’ substantial role as a tumor suppressor in a murine setting. 

The BRCA1 B-complex, characterized by phosphorylated Bach1 at S990, plays a noteworthy role in DNA repair [38]. This modification, catalyzed by CDK during the S phase of the cell cycle, designates Bach1, also known as FANCJ or BRIP1, as a member of the DEAH helicase family, central to the processing of DNA inter-strand crosslinks (ICLs) elicited by chemotherapeutic agents like cisplatin [38,39,40]. The pivotal role of BACH1 extends beyond DNA repair, encompassing its function as an essential tumor suppressor gene. This is evident through clinically relevant mutations identified in breast cancer and the childhood cancer syndrome known as Fanconi anemia [39,40,41]. In a significant study, Peng et al. explored BACH1’s impact on DNA repair and its role in localizing BRCA1 using BACH1-deficient cells. Their findings highlighted BACH1’s dual role, contributing both to DNA repair processes and the localization of BRCA1. Notably, the intensity of BRCA1 foci decreased conspicuously in BACH1-deficient cells following exposure to gamma radiation. This observation suggests that BACH1’s role extends beyond DNA repair facilitation, encompassing the crucial maintenance of BRCA1’s presence within DNA damage foci [42]. 

The interplay between the BRCA1 C-complex and CtIP unfolds in a cell cycle-dependent manner [43]. The interaction, facilitated through phosphorylated CtIP at S327, has many diverse biological functions, including the transcriptional repressor activity of the ZBRK1 C-terminal domain [24,44]. It is worth noting the parallel drawn between CtIP and its yeast counterpart, Sae2, which is instrumental in DNA double-strand break repair through end resection [45,46,47]. This accentuates the regulatory role of the BRCA1 C-complex in directing DNA end resection and influencing the selection of DNA repair pathways, especially evident in chicken DT40 cells [48]. 

Intriguingly, Nakamura et al. utilized the chicken DT40 cell system to reveal additional facets of CtIP’s role [49]. Their investigations revealed CtIP’s essential contribution to generating 3′ single-strand overhangs, a pivotal step in triggering checkpoint responses. Notably, cells bearing a CtIP mutant with compromised BRCA1 binding, referred to as the SA mutant, demonstrated proficiency in homologous recombination (HR) while displaying heightened sensitivity to camptothecin and etoposide, when contrasted with their wild-type counterparts [49]. Building upon these insights, Nakamura et al. proposed that the BRCA1-CtIP complex assumes a fundamental role in eliminating oligonucleotides that are covalently attached to polypeptides at sites of double-strand breaks (DSBs), thereby serving as a facilitator for subsequent and essential DSB repair processes. 

These complex formations underscore the pivotal role played by the C-terminal tandem BRCT domain of BRCA1 in checkpoint control and DNA damage repair. The precise interactions between BRCA1 and its binding partners serve as careful regulators that facilitate various facets of DNA repair, thus emphasizing the vital role that BRCA1 plays in preserving genomic stability.

## 3. BRCA1 Cellular Function

The intricate web of molecular interactions at the cellular level incorporates a range of functions critical for maintaining genomic stability and cellular homeostasis (Figure 2). Among the key players in this complex situation is the BRCA1 protein. Beyond its molecular interactions, BRCA1 is a multifaceted caretaker and gatekeeper, wielding influence over an array of essential cellular processes. This review explores the range of BRCA1’s cellular functions, shedding light on its roles as a guardian against genomic instability, an arbiter of R-loop dynamics, a conductor of cell differentiation, and more. Each facet of BRCA1’s cellular influence contributes to its overarching impact on maintaining cellular health and preventing tumorigenesis. 

In DNA damage signaling, BRCA1 is activated through phosphorylation by ATM/ATR kinases, and it plays a role in the G1–S checkpoint response through indirect mechanisms.BRCA1 interacts with BARD1 and BAP1, participating in protein ubiquitination processes.The BRCA1/ZBRK1/CtIP protein complex is involved in transcriptional regulation.A multitude of BRCA1 protein complexes collectively contribute to the maintenance of genomic stability.

### 3.1. BRCA1 Is a Caretaker and Gate-Keeper

Safeguarding genomic integrity becomes paramount to counteracting the accumulation of mutations that could culminate in cancer [50,51]. Cell cycle checkpoints and DNA repair mechanisms play a pivotal role in upholding genomic stability, with tumor suppressor genes acting as sentinels to thwart cancer development. Vogelstein’s classification delineates tumor suppressors into two distinct roles: caretakers and gatekeepers [52]. Caretaker genes, exemplified by mismatch repair proteins responsible for HNPCC, uphold genome stability by repairing DNA damage. In contrast, gatekeeper genes like RB and p53 regulate cell cycle progression and instigate checkpoints to impede the division of cells harboring compromised DNA.

When evaluating the myriad protein complexes in which BRCA1 participates, BRCA1 emerges as a tumor suppressor gene that operates as both a caretaker and gatekeeper. In its caretaker role, BRCA1 orchestrates a spectrum of DNA repair mechanisms, encompassing homologous recombination repair of DNA double-strand breaks, microhomology-mediated end joining (MMEJ), interstrand crosslink repair, and nucleotide excision repair. Simultaneously, as a gatekeeper, BRCA1 governs cell cycle progression and checkpoint activation. More specifically, BRCA1’s influence spans the regulation of G1/S and G2/M checkpoints, curbing cell cycle advancement in the presence of DNA damage.

Previous discussions underscore the significance of numerous BRCA1-containing complexes—PALB2, A-, B-, C-complexes, and RMN—in various facets of DNA damage repair and signaling pathways. To reiterate, PALB2 links BRCA1 and BRCA2 to facilitate Rad51 loading. The A-complex modulates ubiquitination events at DNA damage sites, the B-complex provides helicase functionality pivotal in interstrand crosslink repair, and the C-complex aids DNA break-end resection. Furthermore, BRCA1 interacts with various transcription factors to modulate gene expression in checkpoint responses. In summary, these interactions critically underpin the importance of BRCA1 and its interactions in maintaining the efficacy of the homologous recombination repair pathway and the activation of cell cycle checkpoints in response to DNA damage.

### 3.2. BRCA1 Prevents R-Loop Formation

A series of compelling observations has revealed a novel role for BRCA1 in safeguarding against the formation of co-transcriptional RNA-DNA hybrid structures known as R-loops [53]. By precisely detecting these R-loops using the monoclonal antibody S9.6, researchers have underscored BRCA1’s role in preventing double-strand breaks (DSBs). 

Previously, the mechanism by which BRCA1 deficiency impacts lineage-specific differentiation was not understood. A study by Zhang et al. explored this mechanism by evaluating the transcription dynamics of breast cancer luminal epithelial cells, which hold special significance due to their involvement in the origin of BRCA1-associated basal-like breast cancer, a clinically relevant subtype [54]. Zhang et al.’s study revealed that R-loops preferentially accumulate in breast luminal epithelial cells of individuals harboring BRCA1 mutations. Notably, a specific R-loop associated with a BRCA1 mutation was pinpointed upstream of the ESR1 gene, responsible for encoding estrogen receptor alpha (ERα) [55]. Expanding on this, the study examined the consequences of BRCA1 knockdown in ERα+ luminal breast cancer cells. This BRCA1 knockdown intensified the presence of R-loops while concurrently diminishing the transcription of neighboring genes. Remarkably, the detrimental effects of BRCA1 depletion on transcription were mitigated by introducing RNase H1—an enzyme adept at dismantling R-loops. This RNase H1 intervention induced a transformative shift in primary breast cells from BRCA1 mutation carriers, prompting the transition from luminal progenitor cells to mature luminal cells [55].

These collective findings shed light on a new dimension of BRCA1’s functionality—mitigating R-loops—which coordinates luminal cells’ specific transcription and differentiation. This mechanism, in turn, emerges as a potential suppressor of BRCA1-associated tumorigenesis. By unraveling the intricate relationship between BRCA1, R-loops, and cellular differentiation, Rong et al.’s study has not only revealed promising avenues for addressing BRCA1-associated breast cancer therapeutically but also deepened our comprehension of the fundamental molecular processes.

### 3.3. BRCA1 Aids in Cellular Differentiation

BRCA1 is predominantly acknowledged for its pivotal role in DNA repair and its significant contribution to cancer prevention. However, a wealth of compelling evidence points toward an additional role for BRCA1, that is, cellular differentiation, wherein a stem cell metamorphoses into a specialized cell type.

Studies dating back to the mid-2000s have revealed BRCA1’s involvement in the regulation of differentiation across various cell types. For instance, Furuta et al. showed that diminishing BRCA1 levels in mammary epithelial cells (MECs) impairs acinus formation while fostering proliferation [56]. Liu et al. underscored the necessity of BRCA1 expression for the differentiation of estrogen receptor (ER)-negative stem/progenitor cells into ER-positive luminal cells within the breast [57]. Years later, Buckley et al. followed up on this subject through chromatin immunoprecipitation and identified a conserved intronic enhancer region within the Notch ligand Jagged-1 (JAG1) gene that is linked to BRCA1 [58].

Recent studies have further explored BRCA1’s role in breast cancer development and its profound impact on cellular differentiation. Ding et al. investigated myoepithelial cells within normal breast tissues of BRCA1 and BRCA2 germline mutation carriers, non-carrier controls, and sporadic ductal carcinoma in situ (DCIS) [59]. This study revealed a significant reduction in the frequency of p63+TCF7+ myoepithelial cells in normal breast tissues of BRCA1 mutation carriers, possibly contributing to their heightened breast cancer risk. Additionally, a reduced fraction of p63+TCF7+ myoepithelial cells was observed in DCIS, hinting at potential links to invasive progression.

Another study examined the interplay between BRCA1 and hypoxia in the context of cancer cell stemness using breast cancer cell lines [60]. This investigation showcased BRCA1’s role in the regulation of cancer stem cell (CSC)-like traits by demonstrating that reintroducing BRCA1 led to a significant decline in CSC-like populations within breast cancer cells. Notably, the study highlighted how hypoxia hampers the differentiation induced by HDAC inhibitors in BRCA1-reconstituted breast cancer cells. This implies that BRCA1 status and tumor hypoxia should be considered as crucial clinical parameters that could influence the efficacy of HDAC inhibitors as therapeutic agents.

Collectively, these findings provide a comprehensive overview of BRCA1’s indispensability in the differentiation of mammary epithelial cells. The absence of BRCA1 could potentially contribute to the persistence of cancer stem cells within breast tumors, thereby driving tumor progression and fostering resistance to treatment. Unraveling BRCA1’s role in cellular differentiation has the potential to reveal new insights into breast cancer development and may pave the way for innovative treatment strategies.

## 4. Lessons Learned from Animal Models

Animal models have been instrumental in enriching our comprehension of the intricate functions of BRCA1 within the landscape of cancer initiation and progression. While in vitro studies and cell cultures contribute valuable insights, animal models provide an avenue to explore the multifaceted role of BRCA1 within the entirety of an organism. This vantage point reveals the dynamic interplay among diverse tissues and biological systems, illuminating the broader repercussions of BRCA1 malfunction.

Mouse models, in particular, have played a pivotal role in showing the contributions of BRCA1 to cancer (reviewed in [61]). Knockout mice devoid of the BRCA1 gene have yielded pivotal revelations, underscoring the indispensability of BRCA1 for embryonic development [62]. The BRCA1Δ11/Δ11 mouse model stands as a compelling example, engineered to harbor a deletion of exon 11 within the BRCA1 gene—an alteration frequently detected in individuals with hereditary breast and ovarian cancer syndrome. Homozygous BRCA1Δ11/Δ11 mice encounter embryonic lethality, a consequence attributed to compromised DNA repair and regulation of the cell cycle [63]. This outcome resoundingly underscores the critical developmental role played by BRCA1. Intriguingly, it is worth noting that these mice still express a BRCA1 variant featuring the N-terminal RING finger and C-terminal BRCT domain, indicating the essential role of the central region harboring DBR (DNA binding) activity in embryogenesis.

Recognizing the challenges posed by constitutive knockout models, researchers have developed inducible knockout models to address tissue-specific BRCA1 deletion. A prominent illustration is the BRCA1^fl/fl^–MMTV-Cre mouse model, where exons 11 within the BRCA1 gene are flanked by loxP sites [64,65,66]. This innovative design enables precise removal of both genes exclusively within mammary gland epithelial cells. The Cre-mediated excision of exon 11 of Brca1 in mouse mammary epithelial cells triggers increased apoptosis and abnormal ductal development. When crossed with the p53^fl/fl^ mouse model, generating BRCA1^fl/fl^, p53^fl/fl^, and MMTV-Cre, wherein exons 11 within the BRCA1 gene and exons 1-6 within the p53 gene are flanked by loxP sites, the mammary glands of nulliparous Brca1/p53-deficient mice display a distinctive accumulation of lateral branches and extensive alveologenesis, a phenomenon typically seen only during pregnancy in wild-type mice. Notably, due to a defect in proteasome-mediated degradation, progesterone receptors, but not estrogen receptors, are overexpressed in mutant mammary epithelial cells [66]. This model underscores the collaborative synergy between BRCA1 and p53 in suppressing tumorigenesis. Moreover, treating Brca1/p53-deficient mice with the progesterone antagonist mifepristone (RU 486) effectively prevents mammary tumorigenesis. These findings highlight the tissue-specific role of the BRCA1 protein and suggest the potential utility of antiprogesterone treatment in the prevention of breast cancer in individuals with BRCA1 mutations [66].

To further illuminate the nuanced functions within the diverse BRCA1 functional domains, serial knockin mutant mice have been generated [17,67,68,69]. These mutants have yielded surprising results, revealing that the E3 ligase activity within the RING finger region is not crucial for BRCA1’s role as a tumor suppressor [17]. Overall, animal models have proven indispensable in elevating our understanding of BRCA1 and its pivotal involvement in the inception and evolution of cancer. Beyond insights, these models furnish a platform to pinpoint potential therapeutic targets and evaluate innovative treatment strategies. Ultimately, this multifaceted approach has the potential to enhance therapeutic interventions and ameliorate the prognosis for individuals grappling with BRCA1-associated cancers.

## 5. Genetic Modifiers for Cancer Incidence

BRCA1 gene mutations increase susceptibility to breast cancer among carriers, but this risk is influenced by a range of factors. Variables such as the age of menarche and menopause, parity and breastfeeding, hormone replacement therapy usage, additional genetic variants, and lifestyle components like diet and exercise all contribute to shaping the breast cancer risk in individuals with BRCA1 mutations. It is important to emphasize that the impact of these modifiers on breast cancer risk in BRCA1 mutation carriers can vary based on age, family history, and the specific type of mutation. Therefore, personalized risk assessment and genetic counseling are highly recommended for individuals with BRCA1 mutations.

In the realm of genetic modifiers, animal models serve a dual purpose. They not only assist in exploring the intricate genetic interactions between BRCA1 and other pathways but also reveal specific biochemical functions of BRCA1 that are crucial for its tumor suppressor activity [70,71,72]. For example, studies have shown that removing 53BP1 can rescue genomic instability in mice expressing a “RING-less” form of BRCA1 [73]. This underscores how the absence of 53BP1 can partially mitigate embryonic lethality, although it does not fully counteract genomic instability in mice with complete BRCA1 knockout.

Recent research has identified potential therapeutic targets for cancers associated with BRCA1 mutations. Activation of NOTCH1, for instance, has been found to mitigate the impact of BRCA1 deficiency, thereby enhancing cell cycle checkpoints and reducing the occurrence of mitotic catastrophe [74]. This discovery presents a new avenue for potentially treating aggressive and therapy-resistant triple-negative breast cancer (TNBC).

Furthermore, a small molecule inhibitor of Polθ polymerase, known as ART558, has demonstrated targeted interference with the major Polθ-mediated DNA repair process [75]. This intervention induces DNA damage and synthetic lethality in tumor cells harboring BRCA1 or BRCA2 mutations. ART558 also synergizes with a PARP inhibitor, augmenting biomarkers associated with single-stranded DNA and synthetic lethality in cells lacking 53BP1. These findings suggest the potential of Polθ inhibitors as a strategy to target DNA repair vulnerabilities in cancer, particularly in tumors characterized by BRCA1 or BRCA2 mutations.

Nacson et al. introduced a unique Brca1CC mouse model featuring a coiled-coil (CC) domain deletion [76]. This region plays a critical role in recruiting BRCA2 to the site of DNA damage through PALB2 [27]. Brca1CC/CC mice are born at low frequencies, and post-natal mice display FA-like abnormalities. The study revealed that the Brca1CC protein is hypomorphic and facilitates DNA end resection, but it is entirely ineffective in RAD51 loading. This further emphasizes the significance of the BRCA1–PALB2–BRCA2 axis in Rad51 loading. The investigation also explored whether Brca1CC mutant proteins, retaining partial activity, could complement each other in the context of BRCA1-associated Fanconi anemia (FA). The findings indicated that Brca1CC and Brca1Δ11 alleles represent separation-of-function mutations, collaborating to provide sufficient homologous recombination (HR) activity crucial for normal development and hematopoiesis. This implies that compound heterozygosity for functionally complementary mutations may confer protection against FA.

Collectively, these studies illuminate the molecular mechanisms through which BRCA1 impacts DNA repair and cancer development. Moreover, they propose promising avenues for targeting DNA repair vulnerabilities within the broader context of cancer treatment.

## 6. Unraveling the Complex Web of Breast Cancer Risk Factors 

Breast cancer poses a substantial global health challenge, affecting women worldwide due to a combination of genetic and environmental factors. To comprehend its origins, we must consider various elements like age, location, reproductive history, and genetic mutations, which collectively contribute to this disease’s complexity. While hereditary genetic mutations are sometimes relevant, environmental factors such as Bisphenol A (BPA) introduce an additional layer of intricacy. BPA, a component prevalent in everyday consumer items from water bottles to food containers, has become omnipresent, with the CDC reporting detectable BPA levels in over 90% of the population. Notably, BPA’s estrogen-disrupting properties raise substantial public health and safety concerns, underscoring the necessity for a comprehensive investigation into its potential repercussions [77].

### 6.1. BRCA1 Deficiency and BPA Response

Jones et al. extensively explore the biological reaction to BPA within the context of BRCA1 status [78]. Genetic mutations or deletions, such as those involving BRCA1, yield distinct biological reactions upon BPA exposure. Notably, BPA is a potent inducer of mammary epithelial cell growth in Brca1^f/f^–MMTV-Cre models. Using immunohistochemistry targeting PCNA, Jones et al. carefully analyzed mammary epithelial cell proliferation in BPA-exposed mice. Their study unequivocally demonstrates that the xenoestrogen BPA acts as a strong mitogen, particularly in mammary epithelial cells with BRCA1 loss. An intriguing facet of their findings is that impaired BRCA1 function amplifies BPA-triggered cell proliferation, as demonstrated in both cell culture and murine models. This intriguing phenomenon underscores the interplay between environmental cues, notably BPA, and the responsiveness of BRCA1-deficient cells, highlighting the role of environmental factors in influencing cellular behavior [78].

### 6.2. BPA’s Estrogenic Behavior

BPA has garnered attention due to its estrogen-like properties, raising concerns about potential links to cancer and degenerative diseases. Elevated estrogen levels have been associated with genomic instability and an increased risk of breast cancer, partially due to estrogen’s interactions with our genome. Estrogen generates R-loops in our DNA, often causing DNA double-strand breaks (DSBs) [79]. These breaks result in critical mutations as cellular DNA repair mechanisms strive to mend them accurately, potentially altering gene behavior or structure, sometimes even rendering proteins useless. Interestingly, estrogen-induced genomic instability is linked to R-loops, co-transcriptional RNA-DNA hybrids, and the induction of DNA replication-dependent DSBs. A considerable number of DSBs triggered by estrogen are dependent on R-loops. This raises concerns about whether BPA, acting as an estrogen disruptor, might induce R-loop-dependent DSB formations and potentially act as a mutagen. Given BPA’s widespread presence in everyday products, comprehending its multifaceted interactions with cellular processes is imperative.

### 6.3. BPA Effects on ER-Positive Cancer

We conducted a pilot study using ER-positive human breast cancer cells to investigate BPA’s impact. Our analysis covered DNA damage responses, cell survival rates (Figure 3), and R-loop formations (Figure 4) in BPA-treated cells. Strikingly, despite encountering arrest in response to BPA, cell numbers increased (Figure 3A,B) alongside elevated γH2AX (Figure 3C,D), indicating DDR pathway activation. We observed a dramatic increase in RNA-DNA hybrids upon BPA treatment (Figure 4B,C). Moreover, RNase H treatment, which specifically removes the RNA strand of an RNA–DNA hybrid (Figure 4A), removed the signal (Figure 4B,C), indicating antibody specificity for RNA–DNA hybrids. The result showed that BPA induces R-loop formation in a dosage-dependent manner. Notably, the ER-positive breast cancer cell line MCF7 exhibited enhanced resistance to 4-hydroxytamoxifen in the presence of BPA doses detectable in certain human populations (Figure 5A,B). These findings reveal parallels between BPA and estrogen, inducing R-loop formation and triggering DNA strand breaks. Alarmingly, BPA exposure amplifies resistance to cancer therapeutics. Notably, the revelation that BPA can induce R-loops is a novel, previously unreported discovery. This suggests BPA’s potential contribution to genomic instability and breast cancer development. By elucidating potential pathways, our results offer insights into BPA’s role in breast cancer progression. We observe how high doses of BPA promote cellular proliferation while inducing DNA double-strand breaks that could trigger cell cycle arrest in ER-positive breast cancer cells. Our findings extend beyond instability and proliferation, touching on therapeutic implications. BPA-exposed ER-positive breast cancer cells exhibit altered sensitivity to chemotherapeutics, potentially engaging in competition with estrogen for receptor binding. This finding underscores acknowledging environmental influences when shaping effective cancer treatment strategies.

Our pilot study encapsulates a comprehensive assessment of BPA’s influence on ER-positive breast cancer cells. We emphasize the critical need to curtail BPA exposure and seek safer alternatives in daily consumer goods by examining its potential to provoke genomic instability and mold proliferation and modulate therapeutic responsiveness. As we navigate the intricate tapestry of interactions between environmental factors like BPA and genetic mutations, we forge a path toward astute preventive measures and treatment paradigms for breast cancer and its associated conditions.

## 7. A Model for the Tissue Specificity of BRCA Genes

Many hypotheses have been formulated to reconcile the conflicting observations surrounding BRCA1’s dual roles in general cellular function and tissue-specific tumor suppressor activity. Remarkably, several animal models featuring BRCA1 mutations have exhibited an elevated incidence of lymphoma. Schaefer and Serrano’s seminal work established a framework for correlating gene mutations with specific tissues, showcased through their analysis of BRCA1—a pivotal gene associated with hereditary breast and ovarian cancer [82,83]. Their study explored the nuanced variations of BRCA1’s role across tissues, driven by factors such as alternative splicing, protein redundancy, and disparities in DNA repair mechanisms. Their exploration further extended to the potential impact of BRCA1 on hormone regulation, including estrogen and progesterone. This investigation shed light on BRCA1’s complex role in breast cancer, emphasizing the importance of identifying its tissue-specific effects to devise optimal prevention and treatment strategies.

With a mounting body of evidence, particularly highlighting BRCA1’s engagement in DNA damage repair, R-loop prevention, and transcriptional repression, we present a perspective that underscores the significance of functional BRCA1 in swiftly resolving R-loops within all bodily tissues under optimal health conditions (Figure 6). Situations can arise in which both the functionality of BRCA1 and its essential binding partner gene are compromised, either due to mutations or exposure to estrogen-like compounds. In such instances, unresolved R-loops emerge, culminating in an escalation of double-strand breaks (DSBs), thus fostering a cascade of additional mutations. These interconnected events may culminate in the development of hormone-dependent tumor formations, further emphasizing the complex interplay of BRCA1 in the context of breast cancer etiology.

## 8. Perspectives

As the discovery of BPA’s pronounced ability to induce R-Loops surfaces, a pertinent avenue of investigation beckons, namely the assessment of other agents that may similarly promote the formation of these DNA structures. Such an endeavor could prove pivotal in proactively minimizing exposure to risk, a particularly crucial concern for individuals carrying the BRCA1 mutation. Notably, the exposure of ER-positive breast cancer cells to BPA ushers in a transformative phenomenon: a discernible reduction in sensitivity to chemotherapeutics. This intriguing alteration hints at a potential tussle for receptor binding with estrogen. In light of these dynamics, a compelling imperative emerges: BPA levels within patients must be meticulously tracked and scrutinized. This vigilance is vital, particularly when administering tamoxifen, where the optimal therapeutic dose might potentially shift based on the patient’s BPA exposure profile. By navigating these intersections, we inch closer to crafting personalized and effective treatment strategies, poised to impact the battle against breast cancer for each patient profoundly.

As we examine R-loop prevention within the context of BRCA1, it is crucial to determine how specific functional domains of BRCA1 contribute to preventive activity. Indeed, the interplay between BRCA1’s functional domains and individual responses to BPA-like compounds warrants meticulous investigation, having the potential to reveal the nuanced patterns that could underpin differing sensitivities.

Another intriguing dimension is the comprehensive analysis of BRCA1 animal models subjected to prolonged exposure to BPA. This ambitious pursuit entails tracking the long-term consequences of BPA exposure within the context of BRCA1 deficiency. By extending the investigative timeline and scrutinizing the potential impact on cancer incidence, we investigate a fundamental hypothesis: Could BPA amplify the tissue-specific tumorigenic potential of BRCA1, catalyzing cancer initiation and progression? The extended study of BRCA1 animal models exposed to BPA can offer critical insights into the dynamics between genetic predisposition, environmental factors, and the development of tissue-specific malignancies. This endeavor deepens our understanding of BRCA1’s many roles and advances our ability to devise targeted interventions to mitigate the interplay between genetic susceptibility and environmental triggers. As we understand how modern chemicals like BPA interact with the complexities of breast cancer etiology, we inch closer to a future where proactive prevention and tailored treatments are key.

In conclusion, our exploration of genetic and environmental factors in breast cancer etiology has revealed a level of complexity that demands comprehensive understanding. The multifaceted role of BRCA1, from its dual functions to its tissue-specific effects, underscores how both inherited mutations and environmental triggers shape the course of this formidable disease. The impact of agents like BPA, with its estrogenic attributes and potential to induce genomic instability, urges us to recognize the need for tailored strategies that encompass both genetic predisposition and external influences.

The journey through BRCA1’s diverse functionalities, from R-loop prevention to hormone regulation, reveals many potential interventions and treatments. Our discussions have illuminated the delicate balance between functional domains within BRCA1 and their varying responses to compounds like BPA. Through meticulous analysis of BRCA1 animal models, coupled with probing investigations into altered chemotherapeutic sensitivity and hormone competition, we move closer to unraveling the intricacies of breast cancer’s development and progression.

In our pursuit of effective preventive measures and innovative treatment approaches, the collaboration between genetic insights and environmental impact is essential. The unraveling of BRCA1’s tissue-specific effects informs the personalized care of affected individuals and allows us to mitigate the influence of environmental agents, reducing their impact, especially for BRCA1 carriers. As we strive to decode the genetic–environmental interplay, we navigate toward a future where tailored interventions harmonize with an individual’s genetic landscape, paving the way to improved outcomes and a brighter outlook for breast cancer patients worldwide.

## Figures and Tables

**Figure 1 ijms-24-14276-f001:**
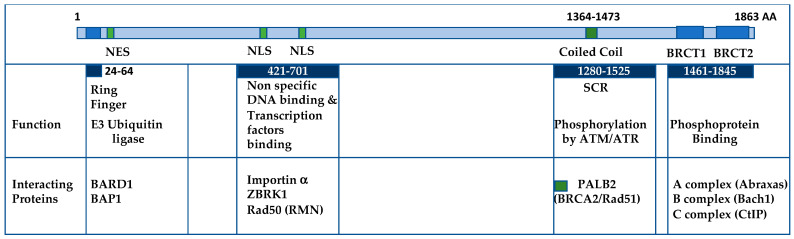
Structural Overview of BRCA1 and Its Functional Domains. The N-terminal region of BRCA1 features the RING finger domain, which, upon forming a complex with BARD1, exhibits E3 ubiquitin ligase activity. This is followed by the NES (nuclear export sequence) responsible for regulating BRCA1’s subcellular localization, along with two NLS (nuclear localization signals) interacting with Importin alpha for nuclear import. The central region of BRCA1 associates with DNA binding activities and interacts with various transcription factors. The coiled-coil domain resides within the SCR (Serine-containing region) and is critical for binding to PALB2, facilitating the recruitment of BRCA2 and Rad51 to sites of DNA damage. The C-terminal tandem BRCT domains participate in phosphoprotein binding, forming the A-, B-, and C-complexes. Key protein binding partners within the BRCT domains include Abraxas, Bach1, and CtIP.

**Figure 2 ijms-24-14276-f002:**
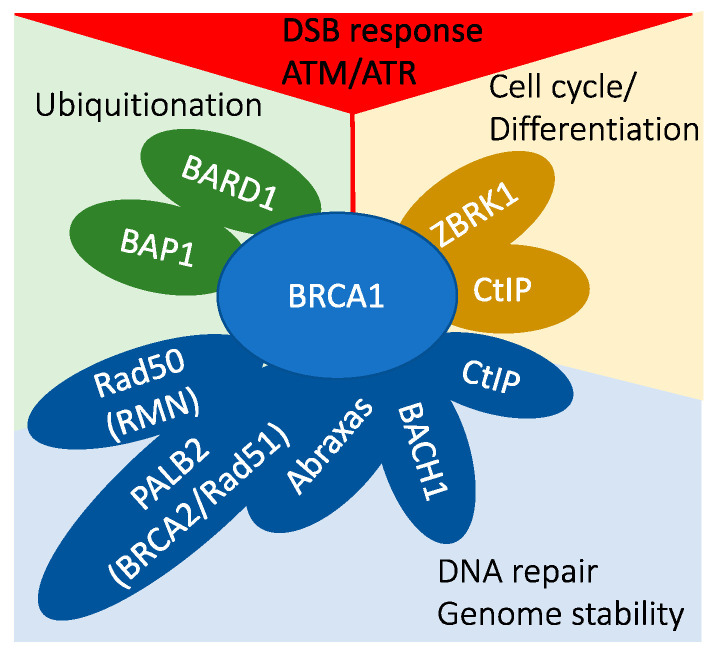
BRCA1 is a central player in various interconnected cellular pathways.

**Figure 3 ijms-24-14276-f003:**
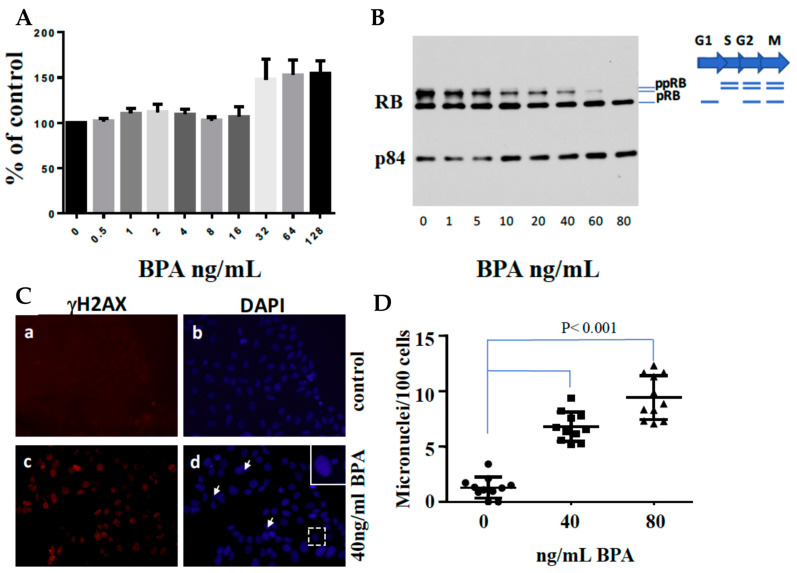
Effects of BPA on Cell Growth, Cell Cycle, DNA Double-Strand Breaks, and Micronuclei Formation in MCF7 Cells. (**A**) MCF7 cells were seeded at 5 × 10^3^ cells/well and incubated for 24 h. Subsequently, a fresh medium containing various doses of BPA was added, and cells were cultured for 96 h. Following this, XTT solution was added for a 4-h incubation at 37 °C. Absorbance was measured at 450 nm using a microplate reader. The values represent the mean ± SEM of six replicates. (**B**) MCF7 cells were treated with different doses of BPA for 72 h. Total protein lysates were extracted and subjected to SDS-PAGE analysis, focusing on RB expression. Protein loading was verified using p84 expression. (**C**) MCF7 cells were cultured on coverslips in 6 cm plates in the presence or absence of 40 ng/mL BPA for 48 h. Cells were fixed and immunostained with anti-gH2AX antibodies (a and c) and counterstained with DAPI (b and d). The arrows point to micronuclei. (**D**) Micronuclei formation following exposure to 0, 40, or 80 ng/mL BPA. MCF7 cells were cultured on coverslips in 6 cm plates with or without BPA for 60 h. Cells were fixed, stained with DAPI, and microscopically analyzed for micronuclei presence (n = 11).

**Figure 4 ijms-24-14276-f004:**
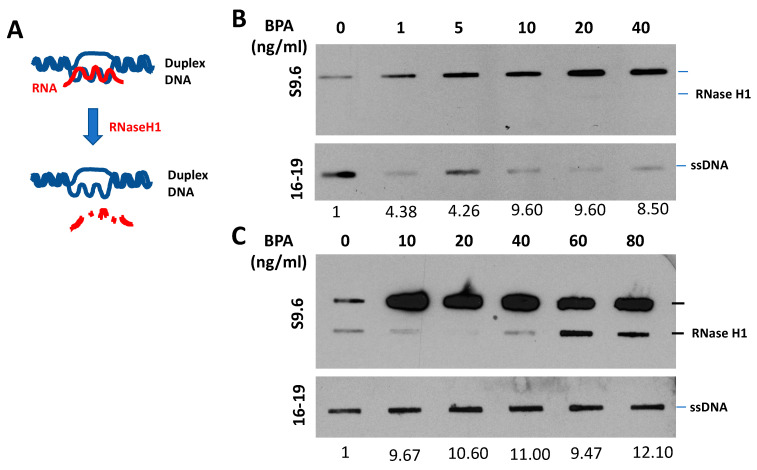
BPA-Induced R-Loop Formation and Detection. (**A**) Diagram illustrating the RNase H1 treatment procedure. (**B**,**C**) Slot blot analysis using S9.6 antibody [80] to detect global RNA–DNA hybrids in MCF7 cells. Cells were treated with various concentrations of BPA (0, 1, 5, 10, 20, 40, 60, 80, and 100 ng/mL) for 48 h. Denatured DNA was detected using a single-strand DNA antibody (16–19). RNase H1 treatment was applied as indicated. Quantitative analysis of nucleic acid bands from slot blot films was carried out using ImageJ software, https://imagej.nih.gov/ij/download.html, accessed on 16 August 2023. Each signal was normalized to ssDNA and presented as a fold increase compared to the control.

**Figure 5 ijms-24-14276-f005:**
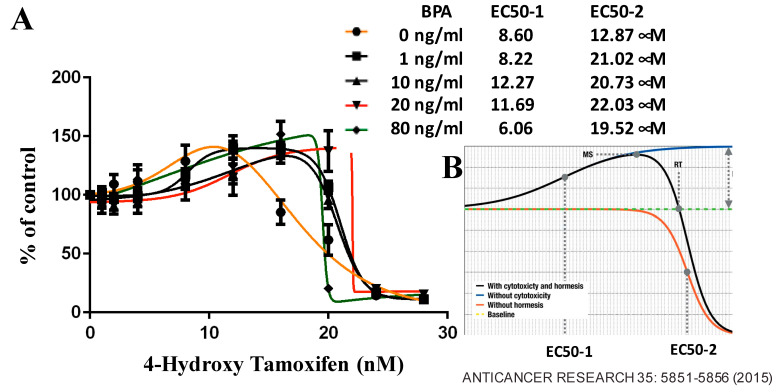
Modulation of Chemo-Sensitivity by BPA in MCF7 Cells. (**A**) MCF7 cells were initially seeded at 5 × 10^3^ cells per well. After 24 h, cells were replenished with 100 µL of fresh medium containing specified doses of 4-hydroxytamoxifen and BPA for 96 h. Subsequently, 50 µL of XTT solution was added to each well and incubated for 4 h at 37 °C. The absorbance at 450 nm was measured using a microplate reader. Growth inhibitory effects were assessed using GraphPad software (https://www.graphpad.com/, accessed on 16 August 2023) to determine the EC50 values. Each data point represents the mean ± SEM of six replicates. (**B**) Representative hormetic dose-response curve [81] illustrates the complex relationship between BPA doses and inhibition of cell growth.

**Figure 6 ijms-24-14276-f006:**
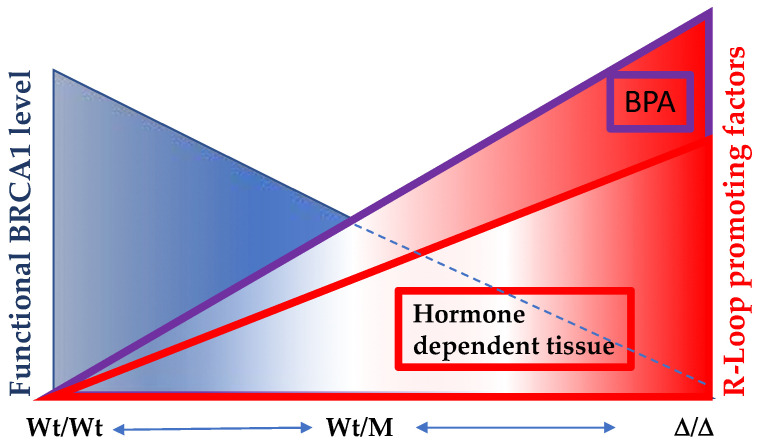
The significance of optimal functional levels of BRCA1 in hormone-dependent tissues extends to the regulation of R-Loop formation. In individuals harboring two copies of wild-type BRCA1 (Wt/Wt), the capacity to resolve R-Loop structures is at its peak. However, among carriers of BRCA1 mutations (Wt/M), the efficacy of R-Loop resolution varies depending on the mutation site. In cases where one copy of the wild-type allele is lost, the remaining allele displays diminished R-Loop resolving activity. Notably, this BRCA1-mediated R-Loop resolution becomes particularly crucial in hormone-dependent tissues, where active transcription sites are susceptible to heightened R-Loop formation. Compounding this scenario is the presence of estrogen-like compounds, such as BPA, which exacerbates R-Loop formation and overwhelms BRCA1’s capacity to counteract it. Consequently, an accumulation of DNA double-strand breaks (DSBs) ensues. Regrettably, BRCA1 also plays a pivotal role in DNA double-strand break repairs, thus creating a challenging situation where compromised BRCA1 function contributes both to increased R-Loop-induced damage and impaired DSB repair processes.

## Data Availability

All data generated or analyzed during the current study are included in this published article.

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
