# Peer review of "BRCA1 the Versatile Defender: Molecular to Environmental Perspectives"

_ijms, 2023, doi:10.3390/ijms241814276_

Round 1
Reviewer 1 Report
Dear Authors,
The Article entitled “ BRCA1 the Versatile Defender: Molecular to Environmental Perspectives” is really an interesting one. The fallowing are few suggestion for a bit betterment of the article
1. Authors mention that with three decades of extensive research efforts, significant advancements have been 61 made in comprehending BRCA1's subtleties. But the claim lacks references.
2. For better understand the ongoing research on the BRCA1, Authors should add the ongoing clinical trial evidences for the support and for better understanding for the readers.
3. The section where authors explain the BRAC1 structural overview is really very interesting. Authors can add a graphical presentation of the Role of BRACA1 in cancer can make the section 2 more interesting and understandable
4. Authors should provide a high-resolution image of figure 2C and 4A, B
5. Authors should mention the ongoing clinical therapy used for the targeting the BRAC1 specifically BPA.
The English language is good but too fancy for a scientific paper
A moderate check of the English grammar is needed. Specifically in the introduction and conclusion section.
Author Response
- Authors mention that with three decades of extensive research efforts, significant advancements have been 61 made in comprehending BRCA1's subtleties. But the claim lacks references.
Ans: Thank you for your valuable feedback. We appreciate your thorough review of our work. We understand your concern regarding the claim lacking references. The references cited throughout the review, beginning with the cloning of BRCA1 (Ref 1), support our discussion. To maintain focus and conciseness in our review, we have chosen to concentrate on providing comprehensive coverage of selected BRCA1 research. Once again, thank you for your thoughtful input, which has contributed to improving our review.
- For better understand the ongoing research on the BRCA1, Authors should add the ongoing clinical trial evidences for the support and for better understanding for the readers.
Ans. Thank you for your suggestion. We agree that adding ongoing clinical trial evidence could enhance the understanding of current BRCA1 research. However, it's important to note that certain aspects, such as the synthetic lethal concept involving PARP inhibitors in BRCA1/2 deficient cancers, are still under investigation. Discussing the multitude of ongoing clinical trials goes beyond the scope of our expertise and the focus of this review. Nevertheless, we appreciate your feedback and have considered your suggestion.
- The section where authors explain the BRAC1 structural overview is really very interesting. Authors can add a graphical presentation of the Role of BRACA1 in cancer can make the section 2 more interesting and understandable
Ans. Thank you for your positive feedback regarding the BRCA1 structural overview section and for your valuable suggestion. As a result, we have incorporated a new Figure 2 at the beginning of Section 3 to enhance engagement and understanding for our readers. Your input is highly appreciated, and we are confident that this addition will enhance the overall quality of our paper.
- Authors should provide a high-resolution image of figure 2C and 4A, B
Ans. Thank you for your suggestion. We have taken your feedback into consideration and replaced the figures as requested. The new figures provide higher resolution and clarity for new Figure 3C, Figure 4, and Figures 5. We believe these updated figures better convey the information in our paper. Your input has been invaluable in improving the quality of our work.
- Authors should mention the ongoing clinical therapy used for the targeting the BRAC1 specifically BPA.
Ans. Thank you for your comment and concern regarding ongoing clinical therapy targeting BRCA1, specifically BPA. As of our knowledge cutoff date, there isn't a clinical therapy specifically targeting BRCA1 involving BPA. We appreciate your intention to bring attention to this potential risk factor, and we fully support the idea of encouraging further research in this area to either rule it in or rule it out as a viable therapeutic approach as we did in the perspective section.
The English language is good but too fancy for a scientific paper
Ans. Thank you for your feedback on the writing style of our paper. We acknowledge that writing style can be a matter of personal choice, and we appreciate your suggestion to defer to the editor's professional judgment in this regard. We trust that the editor will provide guidance to ensure the appropriate tone and style for a scientific paper. Your input is valuable, and we will await the editor's recommendations.
Reviewer 2 Report
The authors state that this review provides an overview of the molecular role of BRCA1 as a tumour suppressor with a focus on breast and ovarian cancers. It also explores potential environmental factors as tissue mutagens that facilitate hormone-related tumourigenesis. The authors concluded that a comprehensive understanding of the functions of BRCA1 is crucial for formulating effective approaches to the diagnosis, prevention and treatment of breast and ovarian cancers. Regarding cancer and the role of environmental factors, the authors stated that Identification of environmental factors contributing to cancer development is equally imperative, as awareness is the basis for proactive prevention and risk reduction.
As a result, the researchers suggested that the co-operation between genetic information and environmental influence comes to the fore in the search for effective preventive measures and innovative therapeutic approaches. In relation to the review topic, the authors also stated that uncovering the tissue-specific effects of BRCA1 informs the personalised care of affected individuals and allows us to mitigate the impact of environmental agents by reducing their effects, especially for BRCA1 carriers.
The authors have given detailed information under 7 main headings and many subheadings on the subject and the subject has been made more understandable with 5 figures. It is a very nice and comprehensive compilation that I think will be useful for studies on the subject.
Author Response
Thank you very much for your positive feedback and kind words about our work. We truly appreciate your support and encouragement. We are delighted to hear that you found our compilation comprehensive and potentially useful for studies on the subject. Your positive feedback motivates us to continue our research in this area.
Reviewer 3 Report
This article provides a comprehensive overview of the multifaceted roles of the BRCA1 protein in breast cancer, particularly its potential interaction with Bisphenol A (BPA) as an environmental factor. It highlights BRCA1's interconnected functions within cells, discusses BPA's impact on breast cancer and chemotherapy sensitivity, and suggests the importance of monitoring BPA levels in patients. The article also proposes studying long-term effects of BPA exposure in BRCA1 animal models to understand the interplay between genetics and environment in cancer development. Overall, this review provides valuable insights and is recommended for publication.
Author Response

(The authors gave the same response as above.)

Round 2
Reviewer 1 Report
No Comments